# Neuromonitoring of the Recurrent Laryngeal Nerve Reduces the Rate of Bilateral Vocal Cord Dysfunction in Planned Bilateral Thyroid Procedures

**DOI:** 10.3390/jcm10040740

**Published:** 2021-02-12

**Authors:** Constantin Smaxwil, Miriam Aleker, Julia Altmeier, Ali Naddaf, Mirjam Busch, Joachim Wagner, Simone Harsch, Oswald Ploner, Andreas Zielke

**Affiliations:** 1Endocrine Center Stuttgart, Department of Endocrine Surgery, Diakonie-Klinikum Stuttgart, 70176 Stuttgart, Germany; smaxwil@diak-stuttgart.de (C.S.); aleker@diak-stuttgart.de (M.A.); julia.altmeier@diak-stuttgart.de (J.A.); naddaf@diak-stuttgart.de (A.N.); buschm@diak-stuttgart.de (M.B.); wagner@diak-stuttgart.de (J.W.); 2Outcomes Research Unit, Endocrine Center Stuttgart, Diakonie-Klinikum Stuttgart, 70176 Stuttgart, Germany; simone.harsch@diak-stuttgart.de; 3Endocrine Center Stuttgart, Department of Endocrinology, Diakonie-Klinikum Stuttgart, 70176 Stuttgart, Germany; ploner@diak-stuttgart.de

**Keywords:** thyroid surgery, vocal cord dysfunction, vocal cord palsy, loss of signal, complications

## Abstract

Purpose: Bilateral vocal cord dysfunction (bVCD) is a rare but feared complication of thyroid surgery. This long term retrospective study determined the effect of intraoperative neuromonitoring (IONM) of the recurrent laryngeal nerve (RLN) during thyroid surgeries with regard to the rate of bVCD and evaluated the frequency as well as the outcome of staged operations. Methods: Retrospective analysis of prospectively documented data (2000–2019) of a tertiary referral centers’ database. IONM started in 2000 and, since 2010, discontinuation of surgery was encouraged in planned bilateral surgeries to prevent bVCD, if non-transient loss of signal (ntLOS) occurred on the first side. Datasets of the most recent 40-month-period were assessed in detail to determine the clinical outcome of unilateral ntLOS in planned bilateral thyroid procedures. Results: Of 22,573 patients, 65 had bVCD (0.288%). The rate of bVCD decreased from 0.44 prior to 2010 to 0.09% after 2010 (*p* < 0.001, Chi2). Case reviews of the most recent 40 months period identified ntLOS in 113/3115 patients (3.6%, 2.2% NAR), of which 40 ntLOS were recorded during a planned bilateral procedure (*n* = 952, 2.1% NAR). Of 21 ntLOS occurring on the first side of the bilateral procedure, 15 procedures were stopped, subtotal contralateral resections were performed, and thyroidectomy was continued in 3 patients respectively, with the use of continuous vagal IONM. Eighteen cases of VCD were documented postop, and all but one patient had a full recovery. Seven patients had staged resections after 1 to 18 months (median 4) after the first procedure. Conclusion: IONM facilitates reduced postoperative bVCD rates. IONM is, therefore, recommendable in planned bilateral procedures. The rate of non-complete bilateral surgery after intraoperative non-transient LOS was 2%.

## 1. Introduction

Intraoperative neuromonitoring (IONM) to the recurrent laryngeal nerve (RLN) is almost always employed during thyroid surgeries in Germany. First introduced in 1996, IONM was rapidly adopted in dedicated centers of thyroid surgery and has become widely accepted in the German surgical community [1,2]. To prevent postsurgical vocal cord dysfunction (VCD), dissection of the recurrent laryngeal nerve is considered gold standard [3,4]. IONM has since received a substantial body of clinical research. It was shown early on that a “loss of signal” (LOS) of the RLN detected by means of IONM carried a substantial likelihood of postoperative vocal cord paralysis [5,6]. Acknowledging its clinical utility, the German Association of Endocrine Surgeons (CAEK) published guidelines for a standardized use, intraoperative trouble shooting and interpretation of IONM as early as 2013 [7]. Owing to the growing scientific evidence, the CAEK-guideline on thyroid surgery for benign conditions encouraged the use of IONM in 2016. Moreover, it suggested that the result of IONM could be used for intraoperative decision making. Already in 2012, a paper suggested to refrain from resection of the contralateral lobe in the case of a LOS of the RLN on the first side of the planned bilateral procedure aiming to prevent bVCD [8].

Bilateral VCD (bVCD) is a rare condition, reported to occur in 0.2–1.2% of thyroid surgeries [5,9,10,11]. Because bVCD is likely to cause significant obstruction of the airway, urgent respiratory therapy and tracheostomy are often needed for relief of symptoms. Although bVCD often is temporary, its impact on the affected individual as well as the health economic burden are significant. Previous reports suggest IONM of the RLN to facilitate reduced rates of bVCD [12,13,14,15]. IONM does not prevent injury to the nerve once occurred, but allows us to document an electromyographic “loss of signal” which, especially if the LOS does not recover during the procedure (i.e., non-transient LOS, ntLOS) correlates to a loss of function of the vocal cord most of the time [12,13,14,15]. It has been suggested that bVCD rates should decrease if a strategy is adopted to consider discontinuation of a bilateral procedure in the event of unilateral ntLOS. Data to support such a concept are scarce [13,15]. Moreover, little is known about the incidence as well as the outcome of staged operations in the event of a halted procedure [13,15,16,17].

We, therefore, analyzed our Thyroid Center Database comprising of a large patient sample over a long period of time. To this end, annual rates of bVCD prior to and after implementing IONM to check the functional integrity of the NLR were produced and the clinical outcome of a change of surgical strategy because of intraoperative ntLOS in planned bilateral thyroid procedures was analyzed in detail in the most recent period.

## 2. Materials and Methods

### 2.1. Study Population

The data presented in this publication represent a single center retrospective analysis of prospectively collected data from January 2000 to December 2019 extracted from the Quality Assurance Registry. The registry has been in use since 2000, recording all thyroid procedures with and without adverse events. Annual reports were produced for internal review. In 2009, the registry was redesigned to comply with requirements of the German Association of Endocrine Surgeons (CAEK) of the German Association of Surgeons. Since 2017, the StuDoQ registry of the DGAVC has been in use. At all times, entries into the registry were done in a prospective fashion. All data computed for this publication were pseudonymized or aggregate non-individual data.

### 2.2. Implementation of Intraoperative Neuromonitoring of the Recurrent Laryngeal Nerve—Reflecting the Evolution of the Use of IONM in Germany

During the 19-year period, 7 endocrine staff surgeons (including 3 changes in staff) were responsible for all of the surgeries reported here, and 4 of the 7 surgeons are still active. Until 2015, with few exemptions, all of these procedures had been total thyroidectomies and bilateral dissection to visualize the recurrent laryngeal nerve was standard. IONM was introduced in 2000, however, it was only incidentally used until 2005, where a broader application was initiated. In 2009, the decision was made to always use IONM to reassure that a dissected recurrent laryngeal nerve was well preserved and functioning. Intermittent IONM comprised stimulation of the vagal and the recurrent laryngeal nerve (RLN), was mandatory and always documented at the end of resection of either thyroid lobe. For the purpose of this study, any loss of signal that did not improve intraoperatively (i.e., non-transient LOS, either global type 2 and/or segmental type 1) and that had occurred after a functioning initial result had been obtained, was considered a potential injury to the RLN. In accordance with the literature, the term “loss of signal” (LOS) was defined and used in any event with a failure of intermittent IONM to elicit contractions of the VC-muscle. Only if LOS persisted (i.e., was not reversible) during the time of the surgical procedure, a “non transient” LOS was recorded. At all times, evolving protocols such as the International Standards Guideline in 2010 or the German Association of Endocrine Surgeons (CAEK) Guideline in 2013 were used for intraoperative trouble shooting and verification of IONM technique [2,7]. Moreover, in the case of ntLOS, dissection of the recurrent laryngeal nerve was done to ensure morphologic integrity of the nerve. 

Since 2010, surgeons were urged to reconsider continuation and/or the extent of surgery in the event of non-transient (nt) LOS on the first side of a planned bilateral procedure. As mentioned above, visual identification of RLN remained the gold standard, and intermitted IONM was used with either needle electrodes or EMG-tube electrodes (as soon as these electrodes became available). All patients had pre- and postoperative VC-tests by direct laryngoscopy on the first postoperative day by an ENT consultant of the ENT-department. Patients with preexisting unilateral or bVCD of all causes were excluded. In this institution, continuous IONM is used in selected cases, such as re-operative surgery and extensive surgery for malignant tumors of the thyroid. However, in order to compare similar endpoints at all times, only non-transient LOS were addressed in this study. To determine the effect of IONM and intraoperative decision-making, bilateral VCD rates prior to 2010 were compared to those after introduction of routine IONM and the option of staged procedures. Differences were tested for significance using Chi2 tests.

### 2.3. Assessing the Incidence of Procedural Change and Staged Operations

To determine the likelihood and the outcome of staged procedures, the clinical results of all thyroid surgeries during the most recent 30 months plus a 10 months follow-up period were evaluated. To this end an unselected, consecutive series of thyroid procedures including re-operative surgery as well as surgeries for malignant thyroid tumors between April 2017 to December 2019 was evaluated using the centers’ StuDoQ-Database. The frequency of ntLOS during planned unilateral and bilateral thyroid procedures, voice outcome, and the frequency as well as the extent of staged procedures was determined. For this analysis, all cases with impaired motility including paresis were recorded as vocal cord dysfunction (VCD) and, therefore, include cases with a complete paresis of the vocal cord (VCP). All patients were followed per protocol with repeat laryngeal examinations by direct laryngoscopy at 6 weeks and 6 months and at 12 months if VCD persisted.

### 2.4. Ethics Approval and Consent to Participate

This study was approved by the Institutional Review Board of the Diakonie-Klinikum Stuttgart and conducted by the Endocrine Centers certified Outcomes Research Unit. All methods were carried out in accordance with the Declaration of Helsinki and the approved guidelines. The data enrolled in this study were obtained from the centers pseudonymized quality assurance database (since 2017 the StuDoQ-Database for certified centers of Endocrine Surgery of the German Association of Surgeons, DGAVC). Ethical committees determined that the data presented in this article do not represent a human participant research study, do not include personal identifying information, and were carried out in accordance with the relevant legislation for the purpose of quality monitoring and quality assurance. Therefore, this analysis did not require informed consent other than the consent obtained prior to entering data into the StuDoQ Database.

## 3. Results

### 3.1. The Incidence of Bilateral Vocal Cord Dysfunction in a High Volume Thyroid Center

From January 2000 until December 2017, a total of 22,573 patients with intact bilateral VC-function underwent first time surgery for benign thyroid conditions. The number of thyroid procedures per year ranged from 892–1438, with an average of some 1200 procedures p.a. IONM had been introduced in 2000, but initially was only used in selected cases. Following a decision to further the application of IONM, its use steadily increased to 50% in 2009. Since 2010, routine use of IONM had been implemented, and IONM applied in basically every patient with a thyroid procedure (Figure 1).

During the entire period, the registry recorded a total of 65 patients to have had postoperative bVCD, for an overall rate of 0.29% (range 0–0.64% per year). The rate of bVCD prior to 2010 averaged 0.44% with a minimum of 0.27% and a maximum of 0.64% per year. With the event of routine IONM (2010–2017) and the option of staged procedures, the average rate of bVCD was recorded to be 0.09% (range 0–0.24% per year). Thus, the average rate of bVCD decreased from 0.44% to 0.09% following introduction of IONM and the option to reconsider the extent of surgery in case of a non-transient LOS (*n* = 56 of 12,664 vs. *n* = 9 of 9909 thyroid procedures, respectively; *p* < 0.001, Chi2). Of all patients with IONM, only one patient with bVCD had no record of any type of LOS at all (<0.01%). From 2017 onwards, there was no case of bVCD on record.

### 3.2. Change of Surgical Procedure in Face of a Potential Impairment of Vocal Cord Function

If an algorithm that takes account of IONM should become surgical practice, patients would need to be informed about the likelihood and the outcome of staged thyroid procedures. We detailed these events in an unselected, consecutive series. From April 2017 to December 2019 (plus a 10 months follow-up period) case reviews were obtained from the centers’ StuDoQ database and comprised of all thyroid procedures, including operations for thyroid cancer (14.9%), recurrent neck surgery (8.2%), and Graves’ disease (7.1%). During 3115 thyroid procedures, non-transient LOS was recorded in 113 cases (3.6%) with a rate of 2.2% per nerve at risk (NAR). Of 113 ntLOS, 100 had a documented VCD on the first postoperative day (83.3%) with a majority of cases (*n* = 89; 74.2%) displaying a complete paralysis of the ipsilateral vocal cord. The rate of any kind of VCD in all patients was 153/3115 (3.0% NAR), and the rate of VCD that had not completely recovered after 12 months was 22/3115. These included individuals that had either persisting VCP (*n* = 7) or were lost to follow up (*n* = 15, last observation carried on forward, LOCF). This produced a rate of persisting VCD of a minimum 0.1% and a maximum 0.4% NAR. There was no case of bVCD (Table 1).

Of the 113 cases with ntLOS, 73 had been recorded during a unilateral procedure. The rate of planned bilateral operations was 952 (30.5%), and the remaining 40 cases of ntLOS were recorded within this group for an overall rate of 4.2%. Of these, 21 occurred on the first side of the procedure. A change of surgical strategy was documented in all of these cases. Procedures were halted in 15 cases, and resection of the contralateral thyroid lobe was not done. In 3 cases, the procedure was continued but restricted to a “limited” sub-total resection leaving a remnant thyroid volume at the ligament of Berry on the contralateral side—as a means to protect exposition of the most vulnerable part of the nerve. In another 3 cases, all of which were thyroid cancer, the contralateral lobe was dissected, making use of continuous vagal IONM (Figure 2). 

Thus, the rate of patients with planned bilateral resections affected by a change of operative strategy was 21 out of 952 (2.2%), and the rate of non-completed resections was 18 out of 21 (1.9%). Of the 18 patients with documented VCD, impaired motility of the VC resolved in all but one after a median of 5 months. Seven patients had the second lobe removed between 1–18 (median 4) months after the primary procedure. The majority were patients with Graves’ disease and differentiated thyroid cancer awaiting radioiodine ablation. Patients with multinodular goiter were less likely to have a second procedure (Table 2).

## 4. Discussion

In 2017, some 70,000 patients underwent surgery of the thyroid gland in Germany [18]. Almost two thirds of these operations addressed both thyroid lobes, and the majority were total thyroidectomies [11]. Recurrent laryngeal nerve (RLN) injury is a feared complication and prevention of postsurgical vocal cord dysfunction (VCD) during thyroid surgery is paramount. However, the true rate of postsurgical unilateral VCD is not precisely known. Based on data from prospective trials, the risk of permanent VCD has been estimated to be around 3% in expert centers [5,6,8]. Health maintenance organizations in Germany estimated the rate of unilateral permanent VCD to be at least 1.5% [16]. A recent prospective analysis in a larger sample found a rate of 10.6% transient and 1.1% permanent VCD, respectively [11].

Although unilateral VCD can already be a significant limitation and occasionally may lead to incapacity to work, bilateral VCD can be much more demanding including the need for tracheostomy [17,19]. Friedrich et al. reported a rate of a permanent bilateral VCD of 0.3% and found the immediate postoperative symptoms of bilateral VCD to be variable—from the rare patient with very little symptoms to severely impaired patients suffering from dysphonia, dyspnea, and extreme stridor [20]. In such cases, temporary ventilation support and permanent tracheotomy are necessary [19]. With respect to the likelihood of VCD, previous research has determined the surgeon to be the most important risk factor. There is cumulative evidence to support the concept, that the individual risk decreases with the number of surgeries per surgeon. Moreover, malignant disease and secondary surgery are additional risk factors [5,16]. Even with careful dissection of the RLN and experience, unilateral and rarely bilateral VCD may also occur. 

With the advent of IONM, hopes were raised that IONM would prevent injury to the nerve, but this has yet to be shown. Whether or not continuous IONM does have the potential to facilitate reduced VCD rates has yet to be established [6,11]. However, permanent loss of neurophysiological nerve signal (LOS) during surgery is highly indicative of an injury to the nerve [12,13,14,15]. This study confirmed that two thirds of patients with nt LOS have postoperative VCD (Table 1). Therefore, IONM may have the potential to assist surgeons, reassuring the functional integrity of the nerve to have been preserved, which is especially important in bilateral thyroid surgeries. The relevance is underscored by the 2.2% rate of ntLOS on the first side of bilateral procedures found in this study. The data of this study show the number of bilateral events to decrease with the use of IONM: From 1 in 200 to less than 1 in a 1000 procedures. On the other hand, and in order to reduce the number of patients affected by bilateral VCD, changes in surgical strategy including incomplete and halted resections are necessary. This concept was first published in 2012 [8,13]. Patients would need to be informed about the likelihood and the outcome of a staged procedure, but until now only a small number of studies have reported on surgical outcomes [8,14,21]. We have found the likelihood of a change of surgical strategy to be around 3% and the rate of incomplete resections to be around 2% in planned bilateral surgeries comparing well to previous reports [14,21]. 

Moreover, a halted procedure in patients with benign nodular thyroid disease allows for a critical appraisal of clinical signs and symptoms after removal of the dominant thyroid lobe—and may not always require a complete thyroidectomy. This contrasts a recent suggestion to start a bilateral procedure on the side of the smaller of the two thyroid lobes—in order to get more space to safely dissect the larger one [22]. We aim to start the procedure on the side of the dominant lobe allowing us to entertain the option of a halted procedure after removal of the most significant lobe. It is, however, noteworthy that the likelihood to halt the procedure or to continue surgery differs between countries. In Germany, most endocrine and thyroid surgeons would be willing to discontinue surgery in case of ntLOS [8], whereas in France, most of the surgeons would continue [23]. It would, however, be reasonable to assume that along with the increasing use of IONM, an increasing acceptance to refrain from bilateral surgery in the event of unilateral ntLOS may be seen [24,25]. Recently, authors from the UK recommend staged thyroidectomies only for “less experienced surgeons” [22]. Although, one should be reminded, however small an experienced surgeons’ rate of postoperative VCD may be, continuing surgery after ntLOS on the first side will always carry a discernable risk of bVCD with the potential of life-threatening consequences.

There are some limitations of this study for critical discussion. This is not a stringent prospective observational trial. It is a report of real-world data from a thyroid surgery registry over a rather large period of time. As such, it reflects the technical evolution of the application and interpretation of IONM. However, as outlined in this report, bVCD is quite rare and prospective studies, until now, have failed to bring forward numbers large enough to sufficiently address the use of IONM in this rare event. Moreover, valid data from prospective multi-institutional registries are currently not available, but may be published in forthcoming years [11].

Another weakness of this study is that the registry only allowed for a correlation of non-transitory LOS with postoperative VCD. Transient LOS or “combined events” were recorded only after 2015 and 2017, respectively. When IONM was introduced and until 2010, these events had yet to be named and to be properly characterized. However, ntLOS is a clear cut and decided endpoint of IONM, that requires no interpretation and, as such, can readily be used for intraoperative decision making. This study has confirmed, as others have, that ntLOS carries a rather high likelihood of postoperative VDP. Nevertheless, the considerable rate of VCD without ntLOS documented in this study indicates the future potential of continuous IONM. Whenever intermittent IONM is used, there is a possibility of missed transient events of impaired signaling which might be a sign of nerve damage. In fact, early experience with continuous IONM indicates that impaired transduction of signal may be used to alter the surgical approach to the NLR and avoid injury. However, this observation awaits confirmation by a PRCCT [6]. 

Finally, this study did not evaluate the relationship between LOS, the extent of a bilateral surgical procedure, the weight of the specimen removed or the results of histopathology. It only addressed the event of unilateral or bilateral VCD. However, from a patient’s perspective this may be the most relevant factor. 

Despite these limitations, this large single institution database from a rather small number of responsible surgeons could minimize the possibility of inconsistency of clinical data, surgical technique, and intraoperative management policy. We are, therefore, quite confident, that our finding of IONM to be a useful application in bilateral thyroid procedures is to be reassured during more elaborate future trials.

In conclusion, we believe this study demonstrated that intraoperative non-transitory LOS mandates reconsideration of the thyroid procedure and that this may change the overall surgical outcome to the better. Postoperative bVCD rates can be reduced by means of IONM and taking account of intraoperative ntLOS. We believe that IONM should be available in every case of a planned bilateral thyroid procedure. We found the likelihood of an incomplete surgical procedure as a result of such intraoperative decision-making to be around 2%. 

## Figures and Tables

**Figure 1 jcm-10-00740-f001:**
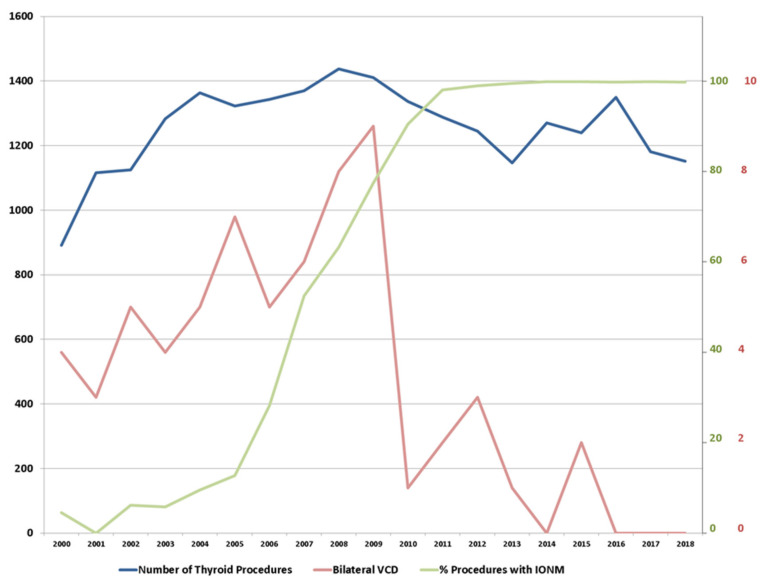
Synopsis of the number of thyroid procedures (scale on the left in black), percentage of procedures with intraoperative neuromonitoring (IONM) (scale on the right in green) and bilateral vocal cord dysfunction (VCD) (scale on the right in red).

**Figure 2 jcm-10-00740-f002:**
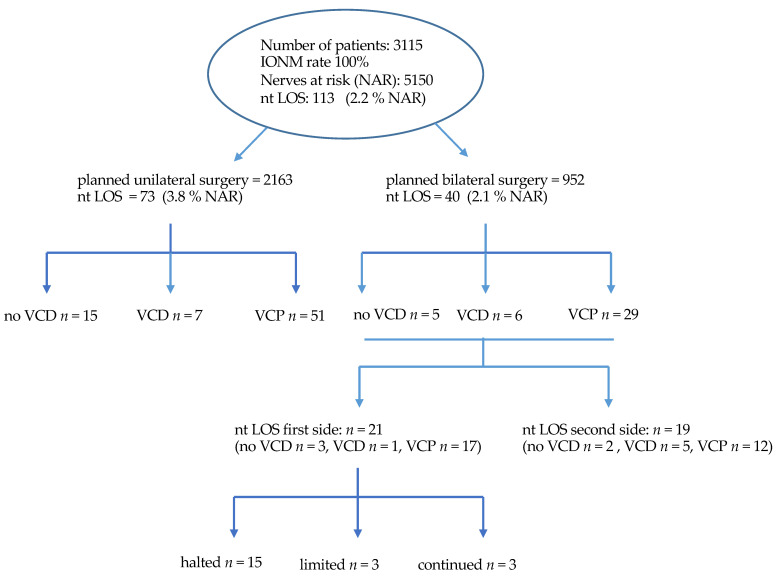
Flow chart depicting number of cases, uni- and bilateral thyroid procedures, events with ntLOS and the respective voice outcome, as well as the extend of thyroid gland resection.

**Table 1 jcm-10-00740-t001:** Synopsis of the indication for surgery, results of the pre- and postop VC assessment, and detailed numbers regarding ntLOS and VCD (April 2017–December 2019, plus 10 months follow up). VCD includes any type of impaired vocal cord function and includes vocal cord paresis (VCP).

All Cases with IONM		3115	100%
	Multinodular Goiter	2173	69.8%
	Graves’ disease	222	7.1%
	Repeat Surgery	256	8.2%
	Thyroid Cancer	464	14.9%
	Planned unilateral surgery	2163	69.4%
	Planned bilateral surgery	952	30.9%
	“Nerves at risk”, NAR	5150	
VCD			
	Preoperative VC-dysfunction (VCD)	36	1.2%
	Preoperative VC-paresis (VCP)	27	0.9%
	Unilateral postop VCD	153	3.0% *
	Unilateral postop VCP	120	2.3% *
	Bilateral postop VCD	0	
ntLOS			
	VCD without nt LOS	53/153	34.6%
	VCP without nt LOS	31/120	25.8%
	Cases with nt LOS	113	
	ntLOS without VCD	20	17.7%
	ntLOS with VCD	100	83.3%
	ntLOS with VCP	89	74.2%
	Persisting VCD, min.	7	0.1% **
	Persisting VCD, max. (15 LOCF)	22	0.4% **

* % NAR; ** % NAR, data presented as a minimum rate and a hypothetical maximum rate including patients either unwilling to take a VC-test or lost to follow up and in whom the last observation of VCD is carried on forward (LOCF).

**Table 2 jcm-10-00740-t002:** Perioperative findings of patients with ntLOS on the 1st side of a planned bilateral procedure: Intraoperative response to LOS (continuation, limited resection, or stopping the procedure), as well as postoperative status of VC function and further treatment.

Preoperative Diagnosis	Response to Nt LOS	VCP (Months)	Further Treatment
Graves’ disease	Stopped procedure	6	No completion, continued follow-up, normal TSH without medication, no sign of Endocrine Orbitopathy, full relief from preoperative symptoms
Bilat. MNG + local discomfort, recurrent goitre	Halted procedure	4	No completion, full relief from symptoms
Bilat. MNG + local discomfort	Halted procedure	Persistent	No completion
Bilat. MNG + local discomfort	Halted procedure	4	Completion thyroidectomy after 4 months
Graves disease	Halted procedure	3	Completion thyroidectomy after 5 months
Bilat. MNG + local discomfort and suspicious nodule	Halted procedure	No VCD	No completion, benign histology, patient reports full relief from preoperative symptoms
Bilat. MNG + local discomfort and suspicious nodule	Halted procedure	4	No completion, benign histology, patient reports full relief from preoperative symptoms
Bilat. MNG, hyperfunctioning adenoma	Halted procedure	6	No completion, normal TSH without medication, patient reports full relief from symptoms
PTC 35 mm, pN0 (0/12)	Continued with cIONM	n.d.	Patient refuses to see ENT specialist, claims full control of voice during follow up visits
Bilat. MNG with local discomfort + primary hyperparathyroidism	Continued with cIONM **	12	** Dunhill procedure, leaving contralateral upper pole, procedure continued because of contralateral parathyroid adenoma and hypercalcaemia
Bilat. MNG + local discomfort	Continued with cIONM **	no VCD	** “Limited” resection of a contralateral nodule close to the isthmus, remnant lobe size: 3 mL
Bilat. MNG + local discomfort	Halted procedure	6	No completion, patient reports full relief from preoperative symptoms
Bilat. MNG + bilat. autonomus nodules	Halted procedure	6	Completion thyroidectomy after 9 months
PTC at multiple sites, 7 mm, pN0 (0/5)	Continued with cIONM	4	Radioiodine treatment
Graves disease	Halted procedure	4	Completion rejected, radioiodine for persisting Graves’ after 8 months
Bilat. MNG + local discomfort and Hashimoto	Halted procedure	3	No completion, patient reports full relief from preoperative symptoms
PTC 8 mm, pN1 (11/37)	Halted procedure	1	Completion thyroidectomy and completion LAD after 2 months, radioiodine therapy
Graves disease	Halted procedure	3	Completion thyroidectomy after 4 months
PTC 18 mm, pN1 (3/8)	Continued with cIONM	3	Radioiodine treatment
Bilat. MNG + local discomfort	Continued with cIONM **	4	** “Limited” contralateral, subtotal resection leaving a remnant of 5 mL
PTC 25 mm, pN (0/16)	Continued with IONM **	no VCD	** Procedure continued after temporary LOS on 1st side, contralateral resection and LAD; nt-LOS of the 1st side (type II) documented at final assessment
Graves disease	Halted procedure	1	Completion thyroidectomy after 2 months

## Data Availability

Restrictions apply to the data presented in this study. Data were obtained from the StuDoQ Thyroid Quality Assurance Registry of the German Association of Surgeons (DGAVC) and may be available to third parties only upon request to the SAVC/DGAVC and decision at the discretion of the DGAVC.

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
