# Peer review of "Neuromonitoring of the Recurrent Laryngeal Nerve Reduces the Rate of Bilateral Vocal Cord Dysfunction in Planned Bilateral Thyroid Procedures"

_jcm, 2021, doi:10.3390/jcm10040740_

Round 1

Reviewer 1 Report

I would like to commend the authors for conducting a large scale study of bVCPs. As they have mentioned in the manuscript, the effect of IONM in preventing bVCPs have yet been published comprehensively. Therefore, the value of this study is of great importance to endocrine surgeons.

I do not believe, however, that the results of this study fully support your conclusion. I think that large control groups such as 1. initial VCP  and continued contralateral thyroidectomy, 2. VCD rates of non-IONM total thyroidectomies must be analyzed in order to justify your conclusion. 

In that regard, I wonder why only the recent 30 months data were evaluated. If you evaluate a broader time period, you may be able to identify more of the 1 group than the three patients you mentioned. 

Also, if the type of LOS was documented for the ntLOS, why weren't any subgroup analyses done accordingly?

Below are some minor check points

-----------------------------------------------------------

Line 34: respectivbely --> respectively

Line 47: dissection of the RLN is considered gold standard -> this expression may be misleading. It could imply that the nerve fibers should be dissected.

Line 55 : 2012 a paper suggested to refrain from... --> do you mean "In 2012 a paper suggested to refrain from.."

Line 100: cervical recurrent nerve --> do you mean recurrent laryngeal nerve?

Line 115: plus a10 months --> plus a 10 months

Line 216 Mayor -> major

Line 231 with the event of --> with the advent of?

Author Response

We are thankful for the benevolent assessment of the paper and the relevant comments.

We would like to respond to the comments and suggestions point by point:

1.As this is a retrospective study, we cannot provide that data. We agree, however, that in a prospective study scenario one data from groups of patients assigned to different modes of IONM-application ex ante would be highly desireable. One could envision a prospective study comparing a group of patients that will have surgery without the use of IONM to groups of patients that will have IONM and different strategies of continuation of surgery depending on the outcome of IONM.

In our current study we tried to do exactly that – looking at different groups of patients that – throughout the course of the almost 20 years – have either received IONM or not. And we have used an endpoint that is well defined and of undisputed clinical significance. And the results of this study will be used for the planning of future research addressing the issues brought forward by the reviewer. In fact, we are part of a CAEK working group that has devised a PRCCT on IONM that is currently under review. A protocol of the future CITY-trial (Continuous vs. Intermittent intraoperative neuromonitoring for reduction of transient recurrent laryngeal nerve palsy in thyroid surgery – a prospective randomized controlled multicenter trial (CITY)) was first presented in 2017 during the German Association of Surgeons Meeting. During the discussion it became apparent, that it is no longer feasible to perform a PRCCT withholding IONM in bilateral thyroid surgery in Germany. IONM is already considered standard of care in Germany and there have been litigations and medical arbitration boards enforcing its general use (Schneider R, Machens A, Lorenz K, Dralle H. Intraoperative nerve monitoring in thyroid surgery-shifting current paradigms. Gland Surgery. 2020 Feb;9(Suppl 2): S120-S128. DOI: 10.21037/gs.2019.11.04.). The protocol of the future CITY trial, which is currently under review for research funding by the Federal Ministry of Education and Research (BMBF), therefore, no longer entertains a No-IONM-Group.

We are aware of the limitations of a retrospective study, and it is therefore, that we have discussed these limitations and referred to the historic numbers of VCD-events. We hope, that the data from our single high-output center may still be worthwhile to be presented to the readers of JCM.

2.With reference to the reviewers’ suggestion for further sub-group analysis of LOS-patients, it appears, that the definitions of LOS may not have been clear enough:
In accordance with the literature, we have defined and used the term “loss of signal” (LOS) in any event with a failure of IONM to elicit contractions of the VC-muscles. If an LOS is recorded, it is rather common that the missing signal (LOS) will recover over time. The timeframe of recovery cannot be predicted in an individual case. Only if LOS is persisting (i.e. not reversible) during the time of the surgical procedure, it turns, per definition, to a “non transient” LOS.
If IONM is done in an intermittent fashion, there can only be two groups of events related to IONM at the end of a procedure:  a group of patients that had a signal at final IONM (no LOS) and those that had no signal during final assessment with IONM at the end of the procedure (non-transient LOS). Because the event of a transient LOS can go completely undetected during intermittent IONM and can only be detected during continuous IONM – this particular subgroup has never been specifically evaluated for intraoperative decision making in trails dealing using intermittent IONM.
We believe, that continuous IONM will close this gap and allow to better study the effect of transient impairment of signal transduction as well as transient LOS on VCD. This is the reason why the CITY trial was designed.

Having been made aware of this potential lack of clarity, we suggest to add the definition of non-transient LOS to the revised manuscript (pg 3, line 94 ff):

In accordance with the literature, the term “loss of signal” (LOS) was defined and used in any event with a failure of intermittent IONM to elicit contractions of the VC-muscles. Only if LOS persisted (i.e. was not reversible) during the time of the surgical procedure, a “non transient” LOS was recorded.

3. The period of the last 40 months was chosen for practical reasons. To evaluate the fate of postoperative VCD, follow up periods of at least one year are necessary. And, given its rarity, it is imperative to have almost complete follow up. Our Endocrine Outcomes Research Facility (EORF), allowing to monitor all of our clinical cases with events, was installed in 2017 at the same time that the national StudDoQ Quality Assurance Registry was implemented in Germany. The rather small number of cases lost to follow up (15/3115; 0,48%) is a strong proof of effectiveness. We simply did not have data of that quality before. And, quite frankly, nobody else has ever reported on such a large number of patients from multicenter studies, let alone a single center.

4. All of the check points mentioned by reviewer 1 have been taken care of in the revised version.

Reviewer 2 Report

Dear Editor,

I read with pleasure the article by C Smaxwil on the reduction of bilateral vocal cord palsy with the use of neuromonitoring.

Bilateral VC palsy is the most catastrophic complication after bilateral thyroidectomy.

Fortunately, it is rare; precisely because of their rarity, it is difficult to do studies that demonstrate the effect of an intervention on the risk of bVCD because there are too few events to analyze.

This study analyzes the results of > 20'000 patients operated on in the last 20 years in a very experienced thyroid center and concludes that the use of neuromonitoring (and the algorithm that suggests to stop surgery after the 1st side in case of LOS) decreases the risk of bVCD.

The study is very interesting and reflects the reality of a very large center.

The limitations are clearly mentioned.

I have 3 comments :

  1. I would suggest to discuss the international recomandations (from the International Neural Monitoring Study Group, (INMSG)) and not only the German ones, for instance
    Randolph G.W., Dralle H., Abdullah H., et al.: Electrophysiologic recurrent laryngeal nerve monitoring during thyroid and parathyroid surgery: international standards guideline statement.121 (suppl 1):S1-S16 2011
    Schneider R., Randolph G.W., Dionigi G., et al.: International neural monitoring study group guideline 2018 part I: Staging bilateral thyroid surgery with monitoring loss of signal.Laryngoscope. 128 (suppl 3):S1-S17 2018
  2. Do the authors have explanation for their high rate of VCD without non-transient LOS ? Did those patients have « transient » LOS ? Did all the surgeons always test the vagus nerve at the end of dissection (“V2” according to the international recomandations) ? It is surprising to see in such a large series that one third of the VCD were not detected by IONM; this fact deserves a more extensive discussion (or perhaps I misunderstood some of the results ?)
  3. In Table 1, I would clarify the fact that VCD include VCP, which is clear in the text “VCD; and therefore include cases with a complete paresis of the vocal cord: VCP » line 122

Author Response

As to comment 1:

Thank you for the suggestion – the international readership of JCM will certainly appreciate to find the evolving international perspective on IONM presented in this paper. Although we had referenced earlier puplications by Diongi as well as Dralle and Schneider, we are happy to include these citations in the revised version. The procedures described in these recommendations, however, are practically identical.

As to comment 2:

Yes, we believe we can offer an explanation. As the reviewer already speculates, there is a possibility of a missed transient LOS ( which can be interpreted as an indicator of transient nerve damage ) whenever intermittent IONM is used. In fact published early experience with continuous IONM (cIONM) allows to speculate, that intermittend LOS as well a impaired transduction of signal may be used to alter the approach ot the NLR in order to avoid injury. However, this observation awaits confirmation by a PRCCT (see comment 1 to reviewer 1).

We were surprised too by the rate of 30% - that is somewhat higher than in other prospective studies. It has prompted us to join the efforts regarding the setting up of a PRCCT using cIONM (see comment to reviewer 1; CITY trial)

Nonetheless, in the revised manuscript we had extended our very brief discussion on pg 8, line 275 ff , highlighting this event and its clinical significance:

Whenever intermittent IONM is used, there is a possibility of missed transient events of impaired signaling which might be a sign of transient nerve damage. In fact, early experience with continuous IONM indicates, that impaired transduction of signal may be used to alter the surgical approach to the NLR and avoid injury. However, this observation awaits confirmation by a PRCCT (21).

As to comment 3:

Thank you, that has been taken care of and in the revised version you should now find this added to the legend of table 1.